# Higher Leukocyte Count Is Associated with Lower Presence of Carotid Lipid-Rich Necrotic Core: A Sub-Study in the Plaque at RISK (PARISK) Study

**DOI:** 10.3390/jcm12041370

**Published:** 2023-02-08

**Authors:** Twan Jowan van Velzen, Jeffrey Stolp, Dianne van Dam-Nolen, Mohamed Kassem, Jeroen Hendrikse, Marianne Eline Kooi, Daniel Bos, Paul J. Nederkoorn

**Affiliations:** 1Department of Neurology, Amsterdam UMC, Location AMC, 1105 AZ Amsterdam, The Netherlands; 2Department of Radiology and Nuclear Medicine, Erasmus MC, University Medical Center Rotterdam, 3015 GD Rotterdam, The Netherlands; 3Department of Radiology and Nuclear Medicine, Maastricht UMC+, 6229 HX Maastricht, The Netherlands; 4CARIM School for Cardiovascular Diseases, Maastricht University, 6200 MD Maastricht, The Netherlands; 5Department of Radiology, UMC Utrecht Brain Center, 3584 CX Utrecht, The Netherlands; 6Department of Epidemiology, Erasmus MC, University Medical Center Rotterdam, 3015 GE Rotterdam, The Netherlands

**Keywords:** carotid stenosis, ischemic stroke, leukocyte, vulnerable plaque, atherosclerosis

## Abstract

Background: Increasing evidence suggests that inflammation inside the vessel wall has a prominent role in atherosclerosis. In carotid atherosclerosis in particular, vulnerable plaque characteristics are strongly linked to an increased stroke risk. An association between leukocytes and plaque characteristics has not been investigated before and could help with gaining knowledge on the role of inflammation in plaque vulnerability, which could contribute to a new target for intervention. In this study, we investigated the association of the leukocyte count with carotid vulnerable plaque characteristics. Methods: All patients from the Plaque At RISK (PARISK) study whom had complete data on their leukocyte count and CTA- and MRI-based plaque characteristics were included. Univariable logistic regression was used to detect associations of the leukocyte count with the separate plaque characteristics (intra-plaque haemorrhage (IPH), lipid-rich-necrotic core (LRNC), thin or ruptured fibrous cap (TRFC), plaque ulceration and plaque calcifications). Subsequently, other known risk factors for stroke were included as covariates in a multivariable logistic regression model. Results: 161 patients were eligible for inclusion in this study. Forty-six (28.6%) of these patients were female with a mean age of 70 [IQR 64–74]. An association was found between a higher leukocyte count and lower prevalence of LRNC (OR 0.818 (95% CI 0.687–0.975)) while adjusting for covariates. No associations were found between the leucocyte count and the presence of IPH, TRFC, plaque ulceration or calcifications. Conclusions: The leukocyte count is inversely associated with the presence of LRNC in the atherosclerotic carotid plaque in patients with a recently symptomatic carotid stenosis. The exact role of leukocytes and inflammation in plaque vulnerability deserves further attention.

## 1. Introduction

Internal carotid artery stenosis (ICAS) is the known cause of stroke in approximately 15% of ischemic stroke patients [1]. More importantly, ICAS is the cause of recurrent stroke in 37% of all patients that suffer from recurrent stroke [2]. In the last decade, imaging-based research on carotid atherosclerotic plaques has identified certain plaque components and morphological characteristics of the plaque that strongly relate to increased risks of stroke and recurrent stroke [3,4]. As such, the presence of intra-plaque hemorrhage (IPH), a large lipid rich necrotic core (LRNC), and a thin or ruptured fibrous cap (TRFC) all appeared to be significant predictors for ischemic stroke on the ipsilateral side of ICAS [3]. Of these vulnerable plaque characteristics, IPH and TRFC show the highest risk for first or recurrent ischemic stroke [5]. For carotid plaque calcifications, conflicting results are shown, with no clear results in the role in stroke prediction [6].

In recent years, the topic of inflammation has gained much interest in the etiological framework of atherosclerosis. In particular, the inflow of leukocytes into the vessel wall [7], which target encapsulated cholesterol, may, in a substantial number of patients, result in a second auto-immune reaction that encompasses the activation of a surplus of macrophages after the endothelial damage has occurred. This activation causes phagocytosis of necrotic cells and damaged cells within the vessel wall [7]. It is thought that this second auto-immune reaction can be one of the causes of plaque vulnerability and subsequent ischemic stroke. Recently, a higher risk of restenosis, after an initial carotid endarterectomy, was shown for patients with higher inflammatory markers [8]. In this study, they investigated several ratios for monocytes, neutrophils, lymphocytes and platelets [8]. Furthermore, both the femoral and carotid arteries were compared through plaque analyses obtained during endarterectomies of femoral or carotid arteries. A higher influence for local inflammation was seen in samples of the carotid plaques through single-cell mapping and comparing to femoral plaques [9]. These recent studies both underline a potential role for (local) inflammation in ICAS.

To our knowledge, no research has been performed to investigate a potential association between leukocyte count and (vulnerable) plaque characteristics in ICAS, while there is a role for leukocytes in the initiation of atherosclerosis itself. The aim of this study, which is a sub-study of the Plaque At RISK study (PARISK), is to investigate if there is such an association between each plaque characteristic individually with the leukocyte count in patients with a symptomatic ICAS of <70% [10]. The PARISK main paper showed that IPH and the total plaque volume can predict recurrent ipsilateral ischemic stroke in patients with ICAS <70% with hazard ratios of 2.12 (95% CI, 1.02–4.44) and 1.07 (1.00–1.15) (per 100 μL increase in volume), respectively [11]. The PARISK main paper emphasizes the high risk that is associated with IPH. The present sub-study could potentially help the further individualization of risk prediction in ICAS patients and a potential better risk model. We hypothesize that an increase in leukocyte count is related to the presence of vulnerable plaque characteristics, out of the hypothesis that inflammatory active plaques are more vulnerable. This inflammatory activity could be reflected in a high leukocyte count and the presence of vulnerable plaque characteristics.

## 2. Materials and Methods

### 2.1. Setting

Patients for this sub-study were derived from the Plaque At RISK (PARISK) study, a diagnostic multicenter cohort study carried out in four academic medical centers in the Netherlands between September 2010 and December 2014. In the PARISK study, patients with an ischemic neurological event in the last three months before inclusion having ipsilateral ICAS < 70% were included. The following exclusion criteria were used in PARISK: any contraindication for MRI or MRI contrast, a clotting disorder, a likely cardiac source of the neurological event, patients in which surgical intervention of the carotid stenosis was planned or performed or patients with multiple comorbidities.

These patients underwent carotid magnetic resonance imaging (MRI) and multidetector-row computed tomography angiography (MDCTA) to assess vulnerable plaque characteristics and general plaque properties. The focus of this sub-study is the association of the individual plaque characteristics, i.e., the presence of IPH, LRNC, TRFC, plaque ulcerations or plaque calcifications (as assessed in PARISK) and leukocyte count. An example of the individual plaque characteristics on CT/MRI can be found in Figure 1, as was published in another PARISK paper [12].

Ethical approval was acquired before the start of this study, and all patients signed informed consent before inclusion in PARISK.

### 2.2. Assessment of Leukocyte Count

Blood leukocytes determined within two weeks from ischemic stroke or TIA onset were post hoc collected in the PARISK patients. Patients with leukocyte count ≥50 × 10^9^/L were excluded, since this hyperleukocytosis could only be explained by a malignancy. Only if data regarding (vulnerable) plaque characteristics and leukocyte count were available were patients eligible for inclusion in this sub-study.

### 2.3. Assessment of Carotid Atherosclerosis

Plaque characteristics were assessed as described in the PARISK protocol [10]. The plaque assessment was performed by blinded assessors and according to PARISK study protocol. In brief, all ipsilateral (symptomatic) carotid arteries were assessed for the presence, and size or volume, of (vulnerable) plaque characteristics. Specifically, for the assessment of IPH, LRNC, TRFC and plaque calcifications, MRI scans were used. For the assessment of plaque ulceration, CTA scans were evaluated.

### 2.4. Assessment of Covariables

Cardiovascular risk factors were assessed using interviews (age, current smoking, medical history of hypertension, hypercholesterolemia, ischemic heart disease or peripheral artery disease, medicine use), sex, kind of event (amaurosis fugax, TIA or ischemic stroke) and physical examination (weight and length). Ischemic heart disease was defined as a clinical diagnosis of myocardial infarction, angina pectoris, or coronary artery bypass grafting or stenting. Peripheral artery disease was defined as typical intermittent claudication complaints or peripheral vascular surgery or amputation because of ischemia. Hypercholesterolemia was defined as a serum total cholesterol of >5.0 mmol/L or the use of lipid-lowering drugs. Body mass index was calculated as the square of length (meters) divided by weight. Age in years at baseline was used. Smoking was scored if the patient reported to smoke during baseline visit. A stroke was defined as TIA if all the symptoms disappeared within 24 h.

### 2.5. Statistical Analyses

We investigated the association of leukocyte count with carotid plaque characteristics using the following strategy. First, univariable logistic regression was used to evaluate the association between the leukocyte count and the separate plaque characteristics (IPH, LRNC, TRFC, plaque ulceration or plaque calcifications). Additionally, we investigated the association for all the covariates individually described above and the plaque characteristics. Second, multivariable logistic regression models were run to investigate the association of the leukocyte count with the separate plaque characteristics, while adjusting for the covariates. The covariates were based on an expert opinion and assumed pathophysiology, since it is unknown which factors could predict the presence of these vulnerable plaque characteristics. SPSS (v26, IBM) was used for the statistical analyses.

## 3. Results

### 3.1. Study Population

Between September 2010 and December 2014, 244 patients were included in the PARISK study, and in 238 of these patients, plaque imaging has been performed. Out of these 238 patients, 161 were eligible for inclusion in this sub-study. In all of the 67 excluded patients, no usable leukocyte count was available. Forty-six patients were female (28.6%), and the median age was 70 (interquartile range [IQR] 64–74). Seventy-six (47.2%) patients suffered from ischemic stroke, seventy-two (44.7%) from a TIA and thirteen (8.1%) from amaurosis fugax. The mean leukocyte count was 7.36 × 10^9^/L (standard deviation [SD] 2.20). IPH was present in 61 (37.9%) patients. LRNC was found in 101 (62.7%), and TRFC was present in 65 (40.4%) of the patients. A calcified plaque was found in 148 (91.9%) of the patients. Finally, an ulcerated plaque was determined in 35 (25.7%) patients. The other baseline characteristics are presented in Table 1, and the imaging biomarkers in Table 2.

### 3.2. Leukocyte Count and Plaque Characteristics

In the univariable regression analyses, we found that higher leukocyte counts were associated with a lower prevalence of LRNC (OR 0.859 (95%CI 0.738–1.000) (*p* = 0.049), but no association was found with the other plaque characteristics (see Table 3). Out of the other variables that were used for univariate regression analyses with LRNC, sex showed a significant association with a *p*-value of 0.002. An overview of the other univariate regression analyses can be found in Table 4.

### 3.3. Multivariable Regression

When multivariable logistic regression for LRNC was performed, with the inclusion of BMI, age, current smoking, sex, hypertension, hypercholesterolemia and the kind of ischemic event as covariates, the association between the leukocyte count and LRNC remained statistically significant with an odds ratio (OR) of 0.818 (95% CI 0.687–0.975, *p* = 0.025). The complete results of the multivariable regression analysis can be found in Table 5.

## 4. Discussion

The precise role for inflammation in plaque vulnerability remains enigmatic. This sub-study in the PARISK study shows an independent inverse association between an increase in leukocyte count (within fourteen days after the ischemic event) and the presence of LRNC on plaque MRI in the sub-acute moment after an ischemic event in patients with symptomatic ICAS < 70%. This finding is not in line with the current hypothesis on the vulnerable plaque. However, to our understanding, this is the first study that investigated the possible association between leukocyte count and vulnerable plaque characteristics.

Since the beginning of this millennium, many papers have been published regarding plaque vulnerability and the specific vulnerable plaque characteristics on MRI in the carotid plaque. Some of these plaque characteristics were associated with a higher risk for either recurrent or first stroke [5,13]. Gupta et al. published one of the first systematic reviews on ischemic stroke risk for some of these vulnerable plaque characteristics with a meta-analysis that showed an OR for ischemic stroke or TIA combined for IPH of 4.59 (95% CI 2.91–7.24), 3.00 (95% CI 1.51–5-95) for LRNC and 5.93 (95% CI 2.65–13.20) for TRFC [13]. Recently, Schindler et al. reported a hazard ratio (HR) of 11.0 for IPH specifically for ipsilateral stroke [5]. These high OR/HR fueled the interest in researching the potential role of inflammation by investigating leukocytes and plaque characteristics.

For a long time, inflammation has been regarded a key factor in the initiation and progression of atherosclerosis. Yet, recently studies focusing on treatment of this inflammatory reaction have published their results [14,15,16]. In the Canakinumab Anti-Inflammatory Thrombosis Outcomes Study (CANTOS), patients with recent myocardial infarction and elevated high sensitivity C-reactive protein were randomized between several doses of canakinumab (a monoclonal antibody against interleukin-1β) and a placebo. They showed a HR of 0.85 (95% CI 0.74–0.98) with *p* = 0.021 for the combined cardiovascular outcome measure (recurrent myocardial infarction, ischemic stroke or other cardiovascular death) [15]. Similar to CANTOS, in the low-dose colchicine for secondary prevention of cardiovascular disease 2 (LoDoCo-2 trial), they tried to reduce the recurrent myocardial infarction, ischemic stroke and other vascular deaths, but with 0.5 g colchicine once daily in patients with chronic coronary disease. They found a HR of 0.69 (95% CI 0.57–0.83) with *p* < 0.001 for the primary outcome [16]. In contrast to canakinumab, colchicine is an old and inexpensive drug that has been used for a long time in the treatment of gout. These studies show promising results for the treatment of inflammation in cardiovascular diseases.

All these studies were performed in patients after myocardial infarction. In stroke patients, the COlchiciNe for prevention of Vascular Inflammation in Non-CardioEmbolic stroke (CONVINCE) trial is currently including patients with recent non-cardio embolic TIA or ischemic stroke. In CONVINCE, the patients are randomized between standard stroke treatment or standard treatment plus low dose colchicine [17]. It will nonetheless take several more years before these results are published.

Even though the studies mentioned above showed a treatment effect towards the prevention of recurrent cerebrovascular events in general and most evidently in a decrease of ischemic stroke cases, no evident blood biomarkers for recurrent stroke have been identified. However, recently, the lipoprotein lipidomics of HDL, LDL and VLDL were used to predict plaque vulnerability and thus (indirectly) stroke risk [18]. The leukocyte count could be considered as a potential biomarker, since we found an inverse association between the leukocyte count as determined in the sub-acute moment and LRNC. However, the found association was the opposite of what was hypothesized, with a lower prevalence for LRNC in ICAS when a higher leukocyte count is found. The role of leukocytes in predicting recurrent cardiovascular events should be tested in the future.

A potential explanation for the inverse relation that was found is the general nature of the leukocyte count as a mix of several types of leukocytes. Fani et al. recently investigated the association between blood immunity markers, both from the innate and adaptive immune system, and vulnerable plaque characteristics in the Rotterdam Study [19]. The Rotterdam Study is a prospective population-based cohort study that started in 1989. Fani et al. used 1602 patients with subclinical carotid atherosclerotic diseases. They found that the leukocytes that are part of the innate immune system (macrophages, granulocytes and platelets) were associated with larger plaques and greater plaque thickness, and that the monocyte-to-lymphocyte ratio was associated with a higher incidence of LRNC. On the other hand, the lymphocytes were associated with smaller carotid plaques and a lower prevalence of IPH. This could be an important reason to assess the leukocyte differential count and calculate the ratios used in their study and make a distinction between the innate and the adaptive immune system in future research. In the investigated setting of the presented study, this association shows the opposite relation as expected, and this could be due to a higher proportion of lymphocytes in the researched population. This could, however, not be investigated, since only the leukocyte count was available.

This study has some limitations. Firstly, the relatively low number of patients included in the analyses. This was mainly due to the fact that this sub-study was not planned when the study was designed and so leukocyte counts were not available in all patients.

Secondly, there is a risk of confounding because of a possibly higher leukocyte count through necrotic brain tissue after the ischemic event. Correcting for this potential confounder was difficult, since NIH Stroke Scale scores (NIHSS) were not available for the included patients. Nonetheless, in this patient population, the estimated effect will probably be minimal, because more than half of the patients suffered from a TIA or amaurosis fugax; and only minor stroke patients could be included (modified Rankin scale ≤3). In these patients, the volume of necrotic brain tissue is generally low, and thus the effect on leukocyte count. Finally, the last limitation of our study is the cross-sectional study design. Due to this study design, we were only able to show the present associations and were unable to investigate the prediction of stroke risk.

The strengths of this study are evident as well. PARISK was a prospective, well-defined, diagnostic cohort study with very complete baseline data. Moreover, it was performed in Dutch comprehensive stroke centers with large experience and specific expertise in MRI plaque imaging. The first strength of this study is the state of the art MR imaging and MDCTA that were performed in these patients [10]. Both imaging modalities used in this sub-study have shown high specificity and sensitivity for detecting the vulnerable plaque characteristics they are used for, and so create high quality data [20,21,22].

Secondly, the patients included in PARISK all suffer from mild to moderate carotid stenosis, with a minimum of 30% measured with ECST criteria. This is the range in which prior research on ICAS showed the least benefit of revascularization through carotid endarterectomy or carotid stenting [23,24]. The investigated patient population could, therefore, be the group in which carotid revascularization is not performed, and thus be of great interest in the prevention of recurrent stroke. Potentially, patients in which the cause of stroke is stated to be cryptogenic, could be caused by vulnerable plaques in the carotid artery, but with a percentage stenosis < 50%. Since these patients were included in the PARISK study, it could give a more accurate depiction of the actual situation within the overall patient population at risk of stroke out of ICAS. This is a valuable inclusion criterion of PARISK.

The final strength of this study is the use of leukocytes that were determined within two weeks around the ischemic event. With a median of one day (IQR 0.00–4.00), it decreases the risk of confounding due to necrotic brain (or eye) tissue. Furthermore, this sub-acute leukocyte count could be a valuable and easy to detect value of the general inflammatory activity within these patients, possibly before the occurrence of the ischemic event as well. This hypothesis deserves future attention.

As stated above, this sub-study found an association between the leukocyte count and LRNC. The direction of the relation is opposite from what was expected. This could be due to an increase in the inflammatory cell count because of the cytokines that are excreted and on the plaque surface that is in contact with the streaming blood. These cytokines could increase the production of leukocytes. It could be hypothesized that plaques that excrete more cytokines are in another phase of plaque progression, and thus have less LRNC compared to plaques that show LRNC more frequently and, in parallel, these patients could show less circulatory leukocytes. This statement is highly hypothetical and deserves further attention in future research, both in clinical as in more translational or fundamental research. At least it could be established that more research into inflammatory biomarkers seems a feasible, but unexplored research field and is highly needed.

Another interesting finding is the negative association between female sex and LRNC in the multivariate logistic regression. It has been shown before that women suffer from auto-immune diseases more frequently, while cardiovascular diseases and atherosclerosis are associated with the male sex [25,26]. Recently, van Dam-Nolen et al. performed a meta-analysis and showed a higher prevalence for all plaque characteristics in males compared to women [27]. The biological or pathophysiological basis for the difference remains unknown, but the hormonal differences during life are thought to be the cause of the difference in the incidence of disease [28]. Since, the PARISK study only found a relation between recurrent stroke and IPH and total plaque volume, the vulnerability of plaques with a LRNC warrants further research. Additionally, the role of sex on the risk for individual plaque characteristics lies out of the scope of this sub-study, but has previously been investigated [29].

Future research should determine if the found association is useful in further risk determination. Additionally, the PARISK study could be used to determine the recurrent stroke risk on the basis of multiple vulnerable plaque characteristics in the form of a new risk model for ICAS. Next to these analyses, a new prospective study should be set up in which patients have blood drawn at pre-specified moments within the (sub-)acute setting (for example on the emergency department, 24 h, three days, one week and two weeks after admission), preferably with the leukocyte differential count, instead of the general leukocyte count that was used in this study. Several studies showed the usefulness of using ratios between some of the components of the differential count. For example, the value of the neutrophil-to-lymphocyte ratio was investigated for predicting the presence of carotid atherosclerosis [30]. This use seems impractical since most patients receive carotid imaging quickly, either during presentation at the emergency department or in the upcoming days, but it could be seen as indirect evidence of the importance of the inflammatory reaction inside the carotid vessel wall. This could be helpful in the initiation of future studies into the treatment of the inflammatory reaction in atherosclerosis. Besides this, the leukocyte differential count has been evaluated for use in predicting outcomes of cardiovascular events, both stroke and myocardial infarction [31,32]. Additionally, a Chinese cohort study showed that borderline elevated cholesterol with a stable or increasing trend has a higher risk for ICAS compared to patients with normal cholesterol levels. This raises the question of whether the limit for cholesterol should be lowered, but first this deserves attention in a clinical setting [33]. Recent research presented a risk model based upon the histological features of endarterectomy samples obtained during surgery, and they were able to show an association between a high score in their risk model and symptomatic ICAS. Whether such a model is feasible in clinical practice should be investigated in future research [34]. Finally, a future prospective study should, next to the leukocyte differential count and other inflammatory biomarkers, at least include carotid MRI and ideally one or several forms of imaging parameters that could detect plaque inflammation, in particular positron emission tomography (PET) CT or dynamic contrast enhanced (DCE-)MRI. The need for more research into plaque vulnerability through these new techniques was recently stressed in a review by Wang et al. [35]. For future research into dynamic MRI techniques, the assistance of deep learning methods has been suggested to aid the determination of plaque vulnerability [36]. Recent research from South Korea showed the presence of IPH in ICAS and in the intra-cerebral circulation, but with quite different prevalence (61.9% vs. 21.4%/16.7, proximal internal carotid vs. basilar/middle cerebral artery). In addition, they showed that progression of IPH was related with progression of the plaque. This could be meaningful in predicting individual plaque risks [37]. In that way, we could learn more on plaque inflammation in relation to time and over different locations throughout the arterial vasculature. It seems valuable to assess the course of the investigated potential biomarkers over time as well.

## 5. Conclusions

In conclusion, this study showed the independent inverse association between the leukocyte count, as determined within fourteen days of the ischemic event on MRI, and the lipid-rich necrotic core. Future research projects should aim at the identification of more (inflammatory) blood biomarkers for both the presence of vulnerable plaque characteristics as well as recurrent stroke in patients with ICAS. This could increase the speed in risk determination and potentially decrease the time until revascularization and so decrease the recurrent stroke risk. Additionally, the assessment of biomarkers could be less expensive than MRI. A future study with both the leukocyte differential count and inflammatory imaging is highly desired to aid in better risk prediction in ICAS patients.

## Figures and Tables

**Figure 1 jcm-12-01370-f001:**
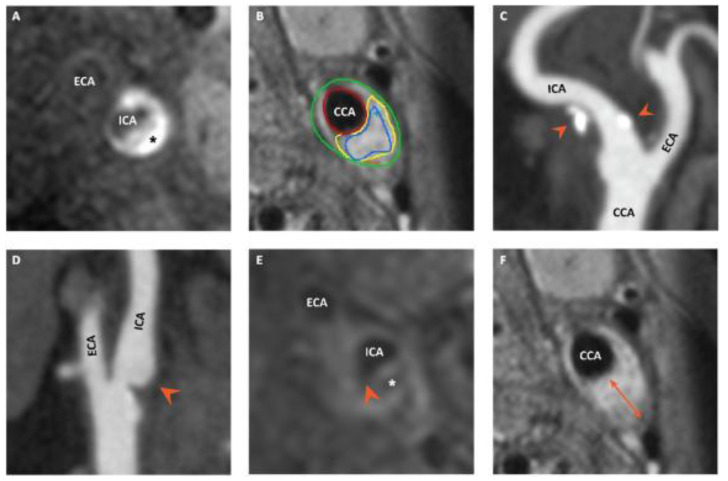
(**A**) Axial view of a 3D T1-weighted fast spoiled gradient echo MR image showing a high-intense signal (asterisk) indicative of intraplaque hemorrhage. (**B**) Axial view of a carotid plaque depicted with 3D T1-weighted pre-contrast quadruple inversion recovery MR, which is delineated to measure different plaque components. The part surrounded with a yellow line is a lipid-rich necrotic core. (**C**) CTA image showing several calcifications (arrow heads) in a carotid plaque of the ICA. (**D**) CTA image with contrast material reaching into a carotid plaque (arrow head), which indicates plaque ulceration. (**E**) Post-contrast T1-weighted fast spin echo MR image, which shows an interrupted signal (arrow head) between the lipid-rich necrotic core (asterisk) and the lumen of the ICA. (**F**) The orange arrow in this 3D T1-weighted pre-contrast quadruple inversion recovery MR image represents the maximum vessel wall area of this carotid plaque. CCA = common carotid artery, ECA = external carotid artery, and ICA = internal carotid artery. These images and legends were previously published by Van Dam-Nolen et al. as part of another PARISK paper [12].

**Table 1 jcm-12-01370-t001:** Baseline characteristics—median values with interquartile range (IQR), means with standard deviation (SD) or number with percentage when appropriate.

Number of Included Patients	161
Baseline Characteristics	
Age, years (median [IQR])	70.00 [64.00, 74.00]
Women (%)	46 (28.6)
Classification event (%)	
TIA (%)	72/161 (44.7)
Stroke (%)	76/161 (47.2)
Amaurosis fugax (%)	13/161 (8.1)
Hypercholesterolemia (%)	81/158 (51.3)
Hypertension (%)	98/161 (60.9)
Diabetes mellitus (%)	35/161 (21.7)
History of TIA or ischemic stroke (%)	43/161(26.7)
History of ischemic heart disease (%)	28/161 (17.4)
History of peripheral artery disease (%)	17/161 (10.6)
Current smoking (%)	38/161 (23.6)
Body mass index (BMI), kg/m^2^ (mean (SD))	26.72/158 (4.11)
Use of cholesterol lowering medication (%)	74/160 (46.2)
Use of antihypertensive drugs (%)	91/161 (56.5)
Use of anti-platelets (%)	60/160(37.5)
Use of anticoagulants (%)	3/161 (1.9)
Leukocyte count	
Leukocyte count × 10^9^/L (mean (SD))	7.36 (2.20)
Time interval between index event and leukocyte count measurement, days (median [IQR])	1.00 [0.00–4.00]

**Table 2 jcm-12-01370-t002:** Imaging biomarkers—number/total number of patients with percentage of total.

Imaging Biomarkers	
Intra-plaque hemorrhage (IPH) (%)	61/161 (37.9)
Lipid-rich-necrotic core (LRNC) (%)	101/161 (62.7)
Thin or ruptured fibrous cap (TRFC) (%)	65/161 (40.4)
Ulcerated plaque (%)	35/136 (25.7)
Plaque calcifications (%)	148/161 (91.9)

**Table 3 jcm-12-01370-t003:** Univariate logistic regression analyses for leukocyte count and individual vulnerable plaque characteristics.

Leukocytes and Plaque Characteristic	OR	95% CI	*p*-Value
IPH	0.910	0.778–1.065	0.242
LRNC	0.859	0.738–1.000	0.049
TRFC	0.891	0.760–1.043	0.152
Plaque ulceration	1.008	0.850–1.195	0.931
Plaque calcifications	1.257	0.888–1.779	0.196

**Table 4 jcm-12-01370-t004:** Univariate logistic regression for LRNC and several baseline variables.

Variable and LRNC	OR	95% CI	*p*-Value
BMI	0.934	0.862–1.012	0.094
Age (year)	0.986	0.946–1.028	0.522
Smoking (current)	1.189	0.555–2.549	0.656
Sex (female)	0.323	0.159–0.655	0.002
Hypertension	1.055	0.548–2.032	0.873
Hypercholesterolemia	1.143	0.601–2.175	0.684
Type of event (ischemic stroke or TIA/amaurosis fugax)	1.428	0.749–2.721	0.279

**Table 5 jcm-12-01370-t005:** Multivariate logistic regression analysis for LRNC with leukocyte count, corrected for other variables.

	*p*-Value	OR	95% CI
BMI	0.096	0.926	0.846–1.014
Age (year)	0.805	1.006	0.959–1.055
Smoking (current)	0.270	1.699	0.663–4.355
Sex (female)	0.001	0.252	0.114–0.556
Hypertension	0.848	0.927	0.430–2.002
Hypercholesterolemia	0.826	1.091	0.502–2.368
Variety of event (ischemic stroke or TIA/amaurosis fugax)	0.217	1.593	0.761–3.337
Leukocyte count	0.025	0.818	0.687–0.975

## Data Availability

The datasets used and/or analyzed during the current study are available from the corresponding author on reasonable request.

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
