# Peer review of "Higher Leukocyte Count Is Associated with Lower Presence of Carotid Lipid-Rich Necrotic Core: A Sub-Study in the Plaque at RISK (PARISK) Study"

_jcm, 2023, doi:10.3390/jcm12041370_

Round 1
Reviewer 1 Report
I would like to congratulate the authors for their work. This is a potentially significant finding because it provides evidence of higher leukocyte counts and lower presence of carotid lipid-rich necrotic core.
Despite the interesting topic of the paper, there are some changes that need to be addressed.
There are only 22 references, which is too few for the original research paper. Furthermore, the authors should include in the analysis other laboratory findings that may have an impact on carotid plaque morphology such as cholesterol (because in Table 1, the authors quantify hypercholesterolemia), triglycerides, HDL, LDL, etc.
It will be very interesting if the authors have a follow-up of the patients, to see if the leukocyte level and the plaque characteristics have an impact on the poor outcome of the patients (TIA, stroke, Mortality), if this information is available.
In the Material and methods section, I strongly suggest that the authors insert some images from MRI or CTA where they highlight the features of the plaque.
I suggest you review and cite the following references:
- https://doi.org/10.3390/ijerph192113934
- https://doi.org/10.3390/ijms232012449
- https://doi.org/10.3390/jcdd9120465
- https://www.mdpi.com/2076-3425/13/1/143
- https://doi.org/10.3390/tomography8040141
- https://doi.org/10.3390/nu14153243
- https://doi.org/10.3390/app12168040
- https://doi.org/10.3390/biom12091192
Author Response
Please see the new cover letter

Reviewer 2 Report
The topic is interesting, and goes to the direction of personalized medicine.
Following are my few suggestions.
I would move ref. #8 from line 72 to line 62, that is where you first cite the PARISK study.
Lines 122-128: usually, text should not repeat what is already reported in tables to spare printed pages, but I understand that you wish to underline the most relevant results, Let’s leave the decision of cancelling these lines or not to the JCM’s Editor.
Table 1: some variables are not defined here, or in the Results section. What do you mean by (history of) ischemic heart disease and peripheral artery disease?
Table 4: female sex, notwithstanding it represents less than a third of the patients, shows a strong negative correlation with LRNC. Can you comment on this? Furthermore, history of TIA/stroke, and ischemic heart / peripheral artery disease all together account for 54.7% of the study population: don’t you think it could be interesting to analyze LRNC in this group of patients?
Author Response
Please see the new cover letter

Round 2
Reviewer 1 Report
no further comments